# Whole Exome Sequencing Facilitates Early Diagnosis of Lesch–Nyhan Syndrome: A Case Series

**DOI:** 10.3390/diagnostics14242809

**Published:** 2024-12-13

**Authors:** Hung-Hsiang Fang, Chung-Lin Lee, Hui-Ju Chen, Chih-Kuang Chuang, Huei-Ching Chiu, Ya-Hui Chang, Yuan-Rong Tu, Yun-Ting Lo, Hsiang-Yu Lin, Shuan-Pei Lin

**Affiliations:** 1Department of Pediatrics, MacKay Memorial Hospital, Taipei 104217, Taiwan; spty871029@hotmail.com (H.-H.F.); clampcage@gmail.com (C.-L.L.); pedhju.4982@mmh.org.tw (H.-J.C.); g880a01@mmh.org.tw (H.-C.C.); wish1001026@gmail.com (Y.-H.C.); 2Department of Pediatrics, Tri-Service General Hospital, National Defense Medical Center, Taipei 114202, Taiwan; 3Institute of Clinical Medicine, National Yang-Ming Chiao-Tung University, Taipei 112304, Taiwan; 4Department of Rare Disease Center, MacKay Memorial Hospital, Taipei 104217, Taiwan; andy11tw.e347@mmh.org.tw; 5Department of Medicine, Mackay Medical College, New Taipei City 252005, Taiwan; 6Department of Childhood Care and Education, Mackay Junior College of Medicine, Nursing and Management, Taipei 112021, Taiwan; 7Department of Medical Research, MacKay Memorial Hospital, Taipei 104217, Taiwan; mmhcck@gmail.com (C.-K.C.); likemaruko@hotmail.com (Y.-R.T.); 8College of Medicine, Fu-Jen Catholic University, Taipei 24205, Taiwan; 9Department of Medical Research, China Medical University Hospital, China Medical University, Taichung 404328, Taiwan; 10Department of Infant and Child Care, National Taipei University of Nursing and Health Sciences, Taipei 11219, Taiwan

**Keywords:** Lesch–Nyhan syndrome, *HPRT1* gene, hyperuricemia, developmental delay, hypotonia, self-mutilation, hypoxanthine–guanine phosphoribosyltransferase

## Abstract

Background: Lesch–Nyhan syndrome (LNS) is a rare X-linked recessive metabolic disorder caused by mutations in the *HPRT1* gene, resulting in hypoxanthine–guanine phosphoribosyltransferase (HPRT) deficiency. Early diagnosis is critical for optimizing management and improving outcomes. This study presents a case series of three Taiwanese patients diagnosed at a single medical center. Methods: Exome sequencing and biochemical testing were used to confirm the diagnoses. Early clinical manifestations, including hyperuricemia, hypotonia, and developmental delay, were documented during the initial stages of the disease. Results: All three patients had hyperuricemia, hypotonia, spasticity, and motor developmental delay. Pathogenic variants in the *HPRT1* gene were identified in two patients, while the third was confirmed by biochemical testing. Two patients had orange-colored crystalline deposits in their diapers, indicative of hyperuricosuria. Self-injurious behavior had not yet developed in two patients due to their young age. Conclusions: Early clinical features such as hyperuricemia, hypotonia, and motor delay may suggest LNS in infancy. Molecular genetic testing, particularly whole exome sequencing, can facilitate an early diagnosis before specific manifestations occur, enabling timely interventions and improving patient outcomes.

## 1. Introduction

Lesch–Nyhan syndrome (LNS) is a rare metabolic disorder first described by Lesch and Nyhan in 1964 [1]. The prevalence of LNS is estimated to range between 1 in 235,000 and 1 in 380,000 live births [2,3,4]. In the UK, the incidence rate has been reported as 0.18 per 100,000 live births [5]. LNS is inherited in an X-linked recessive manner and results from pathogenic variants in the *HPRT1* gene, which encodes the enzyme hypoxanthine–guanine phosphoribosyltransferase (HPRT), a key enzyme in the purine salvage pathway [6,7]. The *HPRT1* gene is located on the X chromosome at Xq26.2-q26.3 and spans approximately 44 kb of DNA. It comprises nine exons and eight introns and encodes a protein consisting of 218 amino acids with a molecular weight of 24.5 kDa [8,9]. Although a single gene is implicated in LNS, over 600 mutations have been identified, resulting in a spectrum of clinical severity due to varying levels of HPRT enzyme deficiency [8]. Males inheriting the defective X chromosome from carrier mothers manifest the disease, while females are typically carriers, although a few may exhibit symptoms if X-inactivation results in the expression of the defective X chromosome [10,11,12]. Genetic testing, specifically rapid exome sequencing, can facilitate an early diagnosis in infants, with detection possible as early as 3 weeks of age [13,14].

Patients with LNS typically present with hyperuricemia and neurodevelopmental abnormalities, including global developmental delay, involuntary movements, and characteristic behavioral abnormalities such as self-injurious behavior [12,15,16]. Effective clinical management can extend life expectancy to 20–40 years [17,18]. Furthermore, emerging research suggests that therapeutic gene correction could offer a promising avenue for future treatment strategies [19]. Affected individuals may not exhibit apparent neurological dysfunction at birth, and developmental delays and neurological signs typically become evident after several months. By 4 months of age, hypotonia and recurrent vomiting are commonly observed. Poor head control is also considered one of the primary initial manifestations [20,21]. Extrapyramidal signs generally appear between 8 and 12 months [22]. While macrocytosis may be detectable as early as 6 months of age, it appears to progress in severity with advancing age [23]. Self-injurious behavior is rarely the initial presentation, but eventually develops in nearly all patients [16,24]. Cognitive impairment and behavioral disturbances typically emerge between 2 and 3 years of age [25].

Self-injurious behavior may manifest as early as 10 months of age, although in some patients it may not develop until adolescence [1,16,26]. Orange-colored crystalline deposits may occasionally be observed in diapers, caused by excess uric acid production [5,27,28]. The aim of this study is to describe Taiwanese patients with LNS from a single center, with a focus on early disease manifestations. By emphasizing the role of whole exome sequencing (WES) in enabling early and precise diagnosis, this study highlights its clinical utility in guiding timely interventions and provides a foundation for future management recommendations.

## 2. Clinical Reports

### 2.1. Case 1

A 1-year-old infant boy presented with hypotonia, intermittent spasticity, and poor head control. He was born at 39 weeks and 4 days with a birth weight of 3414 g. During a clinical evaluation, persistent hypotonia interspersed with episodes of spasticity, intermittent neck hyperextension, and poor head control were noted. In addition, he had mild hypotonia of the extremities, muscle strength graded at 4–5, brisk deep tendon reflexes (+++), and questionable bilateral ankle clonus. His mother noticed orange-colored crystalline sediment in his diaper when he was 4 months old (Figure 1). Developmentally, he showed significant motor delays, being unable to sit or stand independently by 1 year of age.

The family history was notable for a maternal uncle with suspected cerebral palsy who had similar neurological symptoms, including psychomotor delay, self-injurious behavior, and episodes of choking. Unfortunately, as the maternal uncle passed away at the age of 5, it was not possible to obtain genetic or biochemical data to confirm the presence of the familial *HPRT1* mutation. This aspect of the family history is based on the mother’s account, including her descriptions of the maternal uncle’s clinical features, which were consistent with LNS. Cardiac evaluation via echocardiography identified a patent foramen ovale measuring 0.187 cm. Brain magnetic resonance imaging (MRI) revealed the mild enlargement of the subarachnoid spaces, particularly in the bilateral anterior frontal and temporal regions, as well as the mild prominence of the lateral and third ventricles, suggestive of external hydrocephalus. Brain ultrasonography identified bilateral frontal horn cysts, a left subependymal cyst, and subdural fluid collection.

The patient had normal complete blood count, liver function, renal function, and electrolytes and thyroid function tests were within normal limits. Venous blood gas analysis revealed a pH of 7.39 (reference range: 7.35–7.45), a PCO_2_ of 42 mmHg (32–45), a PO_2_ of 32 mmHg (20–49), a HCO_3_^−^ of 25 mmol/L (20–26), and a base excess of 0.3 mmol/L (−2–2). Further metabolic investigations demonstrated an elevated plasma total carnitine level of 81.9 μmol/L (reference range: 30.5–75.9), with a free carnitine level of 55.6 μmol/L (23.6–65.4) and bound carnitine level of 26.3 μmol/L (1.0–16.8). In addition, creatine kinase was elevated at 376 U/L (30–223), and he had hyperuricemia with a serum uric acid level of 10.2 mg/dL (reference range: 3.5–7.2). Based on these findings, advanced genetic counseling and further diagnostic evaluations were recommended.

Molecular genetic testing was performed using the Illumina NovaSeq X Plus/6000 platform (Illumina, Inc., San Diego, CA, USA) with an average sequencing depth of 151× and a 20× coverage exceeding 99%. Variants were analyzed using the DRAGEN v4.2.6 bioinformatics pipeline, and annotations were derived from databases including ClinVar (2023-10), OMIM (2023-10), and gnomAD (v2.1.1, exomes and mitochondria). Functional impact predictions were conducted using PROVEAN, SIFT, and SpliceAI.

WES identified a splice site acceptor mutation at chromosome X position 134498607 (ChrX:134498607), specifically NM_000194.3

c.533-1G>C in the *HPRT1* gene. Follow-up Sanger sequencing was conducted on the proband’s family members, and the results confirmed that both the mother and maternal grandmother were carriers of the *HPRT1* c.533-1G>C variant in a heterozygous state. LNS was diagnosed at the age of 10 months, and he was initiated on pharmacological management including allopurinol, clonazepam, and carbidopa/levodopa. The family also considered hematopoietic stem cell transplantation as a potential therapeutic option.

### 2.2. Case 2

A male infant, aged 1 year and 2 months, born at a gestational age of 39 weeks and 3 days via Cesarean section due to placenta previa, presented with multiple medical complications. His birth weight was 3850 g, and his birth history was G1P1. Postnatally, he exhibited poor weight gain, prompting further evaluation. The nephrologist suspected renal tubular acidosis type I and nephrocalcinosis, leading to the initiation of sodium bicarbonate supplementation. The family reported observing occasional yellowish crystals in his urine and also poor sleep quality with frequent startle reflexes.

At 4 months of age, he developed seizure activity, necessitating multiple hospital admissions. The seizures were managed with antiepileptic drugs, including levetiracetam and clonazepam. Associated symptoms included metabolic acidosis, poor appetite, and frequent regurgitation. Hyperuricemia was noted, raising the suspicion of LNS. A physical examination revealed hypotonia with intermittent spasticity and hypermobile joints. Brain ultrasonography and echocardiography were unremarkable; however, renal ultrasound at 1 year and 1 month revealed bilateral nephrocalcinosis and mild left hydronephrosis. Electroencephalography performed at 1 year and 2 months showed normal findings. Involuntary movements were noted during periods of excitement, and he began treatment with S-adenosylmethionine at 1 year and 3 months. Metabolic investigations during early infancy revealed abnormalities in renal function and metabolic acidosis. Further metabolic investigations demonstrated a reduced plasma total carnitine level of 39.6 μmol/L (reference range: 45.7–63.2), with a free carnitine level of 33.4 μmol/L (37.0–50.5) and bound carnitine level of 6.2 μmol/L (4.8–16.5). Urine organic acid analysis showed elevations in acetoacetic acid, vanillic acid, and sebacic acid. A urinary Benedict’s test was positive, and hyperuricemia was noted with a serum uric acid level of 17.3 mg/dL (reference range: 4.4–7.6). The urine urate-to-creatinine ratio was significantly elevated at 4.0 (111.2/27.6).

Genetic testing via the Sanger sequencing of *HPRT1* revealed a hemizygous guanine duplication at nucleotide position 212 (c.212dupG), resulting in a frameshift mutation and premature termination codon at amino acid position 71 (p.Gly71fs*3). This mutation produces a truncated and likely non-functional protein, consistent with the diagnosis of LNS. The diagnosis was confirmed at 1 year and 3 months of age. According to medical records, treatment included febuxostat, potassium citrate, S-adenosylmethionine, and diazepam. Unfortunately, he was lost to follow-up at 1 year and 6 months of age.

### 2.3. Case 3

This male patient presented to our outpatient department at the age of 4 years. He was born via Cesarean section at a gestational age of 39 weeks, with a birth weight of 2500 g and birth history of G1P1. Clinically, had features reminiscent of cerebral palsy. A physical examination revealed dystonia, spasticity, bilateral ankle clonus, disuse muscle atrophy, and joint contractures involving both hands and ankles. A self-inflicted wound on the lower lip was observed. Hypotonia had been noted between the ages of 1 and 3 years. By the age of 5, he was able to ambulate with the assistance of a walker, but he developed hypertonia by 6 years of age. He had previously undergone a surgical intervention for C1–C2 subluxation. At the age of 10, due to respiratory failure, a tracheostomy was performed. Developmental milestones were delayed throughout infancy and early childhood. Brain MRI identified a focal bony fragment at the odontoid process with a well-defined border, suggestive of either os odontoideum or an old fracture of the odontoid process, with the suspicion of focal narrowing of the upper cervical cord on the sagittal section. Additional findings included left-sided mastoiditis and bilateral sphenoid sinusitis, as well as a relatively brachycephalic skull configuration. Renal ultrasonography revealed bilateral nephrolithiasis with multiple stones in both kidneys, and a staghorn calculus was suspected in the right kidney. Non-contrast computed tomography of the abdomen confirmed the presence of bilateral renal stones, bladder stones, and a suspected stone in the left distal ureter.

Laboratory investigations showed an elevated serum uric acid level (12.1 mg/dL) and macrocytic anemia. Further testing at the Baylor College of Medicine in the USA revealed undetectable HPRT activity (0 nmol/min/mg protein; control range: 1.0–3.4 nmol/min/mg protein; affected control: 0.0 nmol/min/mg protein). Based on these findings, a diagnosis of LNS was confirmed when the patient was 12 years old. He was treated with allopurinol, folic acid, potassium citrate, S-adenosyl methionine, and piracetam during the therapeutic course. Unfortunately, the patient was lost to follow-up at the age of 24.

## 3. Discussion

Currently, the diagnosis of LNS mainly relies on molecular genetic testing and/or low HPRT enzyme activity. In most patients, the disease is caused by a hemizygous pathogenic variant in the *HPRT1* gene in a male proband. Due to the potential lack of early clinical manifestations, hyperuricemia and hyperuricosuria may be among the first indicators of the disease. Some patients may present with normal serum uric acid levels but an elevated urine uric acid-to-creatinine ratio greater than 2 [29]. Due to the overproduction of uric acid, hyperuricemia or hyperuricosuria may manifest early in life. Orange-colored crystalline deposits in diapers and uric acid sediment in the kidneys can be observed as early as a few weeks of age [22]. In our series, all of the patients had hyperuricemia, with orange-colored crystalline deposits observed in the diapers of two patients during the first few months of life. In addition, hypotonia was observed in all patients as early as 4 to 6 months of age, accompanied by poor head control. Intermittent spasticity and hypotonia were frequently noted in subsequent evaluations. All three patients had developmental delays, with a particular emphasis on delayed motor function. None were able to sit independently, crawl, or walk without assistance. In comparison to previous studies, all three of our patients presented with hyperuricemia or hyperuricosuria, hypotonia, spasticity, and developmental delays [20]. These features may serve as early indicators of LNS during infancy. Self-injurious behavior typically emerges between 2 and 3 years of age; however, there have been reports of patients presenting as early as 10 months of age. Self-mutilation was not observed in two of our patients, possibly due to their young age. Although macrocytic erythrocytes are present in 81–92% of individuals with LNS or its neurological variants, macrocytic anemia was observed only in the older patient in this study, suggesting that this symptom may emerge in the later stages of disease progression [23]. While cognitive impairment may be a feature in all patients, it is challenging to accurately assess cognitive function during infancy. The clinical characteristics of our patients highlight the distinctive features of LNS (Table 1).

Molecular genetic testing, specifically the sequencing of the *HPRT1* gene or comprehensive genomic testing, offers a precise diagnostic tool for LNS. Biochemical testing, such as assessing HPRT enzyme activity, is an alternative method to confirm the diagnosis. However, an increasing number of patients are now being diagnosed through next-generation sequencing, which is becoming more prevalent than enzyme activity testing. One of our patients received WES when he was 9 months old, showing that a diagnosis can be made before all of the manifestations occur. Due to the rare incidence of LNS, the three cases presented in this study were selected based on their diagnosis at our hospital and the availability of complete medical documentation. These cases were identified through different diagnostic methods, including biochemical testing and molecular genetic testing, which allowed us to compare traditional approaches with newer techniques such as WES. Timely identification allows for the implementation of appropriate physical interventions and preventive measures before the onset of irreversible manifestations. Moreover, families benefit from genetic counseling and psychosocial support, which are crucial in mitigating long-term challenges associated with the disorder. However, while WES has proven to be a powerful tool in identifying pathogenic variants within coding regions, it has certain limitations that warrant consideration. WES is unable to detect mutations located in non-coding regions of the genome, including regulatory elements or deep intronic regions that may affect splicing. Additionally, WES is less effective in identifying structural variants, such as large deletions, duplications, or rearrangements, as well as copy number variations (CNVs), which can contribute to the pathogenesis of LNS in rare cases [30,31]. Complementary diagnostic techniques, such as whole genome sequencing (WGS) or chromosomal microarray analysis, may be necessary for cases where WES results are inconclusive. Comprehensive evaluation and management led to improved outcomes and a slower progression of the disease in our patients (Table 2). In summary, hyperuricemia or hyperuricosuria, hypotonia, spasticity, and developmental delay may present in early infancy in patients with LNS. Our findings suggest that clinicians should consider LNS in patients presenting with these early manifestations.

## Figures and Tables

**Figure 1 diagnostics-14-02809-f001:**
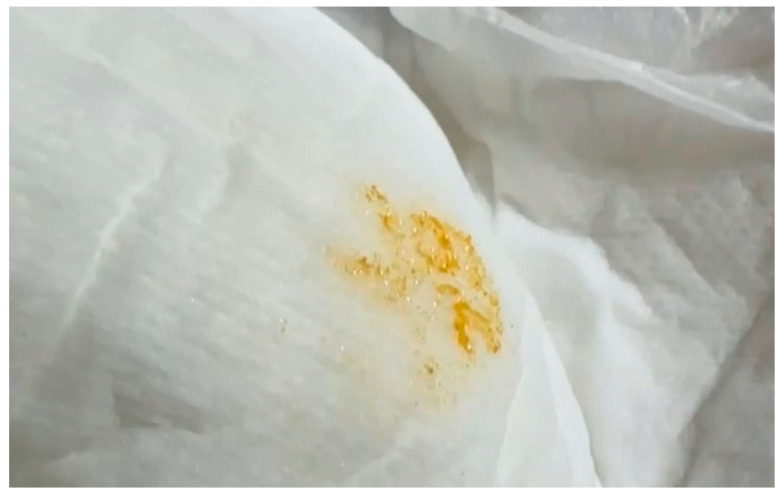
Orange-colored crystalline sediment in the diaper was noted when the baby (case 1) was 4 months old. The sediment is indicative of hyperuricosuria, a key early manifestation of LNS.

**Table 1 diagnostics-14-02809-t001:** Clinical characteristics of three patients with Lesch–Nyhan Syndrome at our hospital.

Patients	Case 1	Case 2	Case 3
Sex	male	male	male
Year of birth	2023	2013	1995
Age at diagnosis	10M	1Y3M	12Y
Diagnostic strategy	Whole exome sequencing: c.533-1G>C/Hemizygote	*HPRT1* gene sequencing: c.212dupG, p.Gly71fs*3/Hemizygote	Hypoxanthine phosphoribosyl- transferase activity: 0 nmol/min/mg protein
Hyperuricemia	10.2 mg/dL	17.3 mg/dL	12.1 mg/dL
Nephrolithiasis	NA	+	+
Hypotonia/ dystonia	+	+	+
Spasticity	+	+	+
Clonus	+	+	+
Cerebral palsy	-	-	+
Delayed motor skills	+	+	+
Intellectual disability	-	-	+
Seizure	-	+	+
Self-mutilation	-	-	+
Gouty arthritis	-	-	-
Macrocytic anemia	-	-	+

NA: no assessment, +: present; -: absent.

**Table 2 diagnostics-14-02809-t002:** Clinical evaluation and management of three patients with Lesch–Nyhan syndrome at our hospital.

Patients	Case 1	Case 2	Case 3
Neuroimaging	Brain MRI: mild enlargement of subarachnoid space most pronounced in bilateral anterior frontal and temporal convexities as well as mild prominence of lateral and third ventricles likely external hydrocephalus at this age. Brain ultrasonography: 1. Frontal horn cyst, bilateral. 2. Subependymal cyst, left 3. Subdural fluid collection	Brain ultrasonography: negative	Brain MRI: negative
Kidney ultrasonography	NA	Kidney ultrasonography: 1. bilateral nephrocalcinosis; 2. left mild hydronephrosis	Kidney ultrasonography: 1. Right small kidney; 2. Bilateral multiple stones in both kidneys (largest in right pelvis: 19 mm, left pelvis: 19 mm); 3. Suspected staghorn stones, bilateral
Treatment	Allopurinol, clonazepam, carbidopa/levodopa	Febuxostat plus potassium citrate, S-adenosyl methionine, diazepam	allopurinol, folic acid, potassium citrate, S-adenosyl methionine, Piracetam

NA: no assessment.

## Data Availability

The original data and findings obtained during this study are fully reported in the manuscript. The corresponding authors can be contacted for additional data or information inquiries.

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
