# Peer review of "Whole Exome Sequencing Facilitates Early Diagnosis of Lesch–Nyhan Syndrome: A Case Series"

_diagnostics, 2024, doi:10.3390/diagnostics14242809_

Round 1

Reviewer 1 Report

Comments and Suggestions for Authors

The manuscript of “Whole Exome Sequencing Facilitates Early Diagnosis of Lesch-Nyhan Syndrome: Case Series and Review of the 3 Literature” by Hung-Hsiang Fang et al. reported three cases of Lesch-Nyhan Syndrome (LNS). The authors thoroughly describe the clinical characteristics of these patients, providing comprehensive knowledge on the diagnosis and clinical management of LNS. Here are some comments for the authors:

1.                The study primarily focuses on three cases rather than a broad literature review. Therefore, it is suggested to revise the title to better reflect the content, possibly removing "review of the literature."

2.                This manuscript emphasizes the benefit of early diagnosis through molecular genetic testing for timely interventions. However, it should be noted that there is no effective medicine for LNS currently. In Line 61 on page 2, the statement "With effective clinical management, the life expectancy of patients with LNS can extend to 20–40 years" should be carefully reconsidered, as it cites older publications. More recent references should be provided if available.

3.                Since the three cases did not follow-up the outcome of pharmacological management, how to clarify “improving patient outcomes”.

4.                Regarding Case 1, it would be helpful to include genotype of HPRT1 for maternal uncle? Did he confirm the familiar HPRT1 mutation? 

Reviewer 2 Report

Comments and Suggestions for Authors

The manuscript provides a clear and detailed case series on Lesch-Nyhan Syndrome (LNS), emphasizing early diagnosis using whole exome sequencing (WES). The clinical and genetic descriptions are well-documented, with illustrative examples and patient data. However, some areas can be improved for clarity and impact. Below are specific comments:

1.      The introduction section can be improved: Expand the introduction to highlight the diagnostic gaps that WES addresses, supported by specific examples from recent literature.

2.      The incidence statistics for LNS provided in references 2, 3, and 4 (published in 1978, 2009, and 2011) are outdated, particularly given the advancements in molecular genetics assays in recent years. The increased accessibility and use of techniques such as whole-exome sequencing and next-generation sequencing may have significantly improved diagnostic accuracy, potentially altering the reported incidence rates. To ensure the manuscript reflects the current understanding, it is recommended to include more recent references or studies that utilize modern genetic diagnostic tools to report LNS incidence. This would strengthen the relevance and accuracy of the epidemiological context presented.

3.      The reference cited for the life expectancy of patients with LNS (reference 15, published in 1993) is significantly outdated. With advancements in molecular genetic diagnostics, early detection, and improved clinical management strategies, more recent data may provide a more accurate and updated understanding of life expectancy in LNS patients. It is recommended to incorporate more recent studies to reflect the impact of modern interventions on patient outcomes and survival rates.

4.      The justification for using WES in the article is valid but could be expanded with a discussion of its limitations, especially regarding its inability to detect mutations in non-coding regions of the genome, as well as structural variants and copy number variations, which are known to contribute to LNS in rare cases.

5.      Clarify the selection criteria for the cases and discuss whether they are typical or atypical of broader LNS presentations. The study does not explicitly state how these three cases were selected or whether they are representative of the general LNS patient population.

6.      Details on the sequencing protocols (e.g., platform used, depth of sequencing, bioinformatics pipeline), methods for the clinical and neurological assessments are missing.

7.      The statement that early WES diagnosis improves outcomes is speculative, as the manuscript does not include follow-up data on this patient outcomes.

  1. Minor grammatical errors in lines like "With effective clinical management, the life expectancy...can extend to 20–40 years" could be refined for conciseness.
  2. Figure 1 lacks clear annotation or explanation within the figure legend.

10.  Abstract, remove duplication: This study presents a case series of three Taiwanese patients diagnosed at a single medical 26 center. Methods This case series involved three male patients with LNS.

11.  Introduction, the aim does not correspond to the title (no WES mentioned)

12.  Lines 234-235, No information on Helsinki declaration revision year

Round 2

Reviewer 2 Report

Comments and Suggestions for Authors

the authors thoroughly revised the text according to my comments. I have no more comments on the manuscript